# Prognostic and Predictive Value of Tumor-Infiltrating Immune Cells in Urothelial Cancer of the Bladder

**DOI:** 10.3390/cancers12092692

**Published:** 2020-09-21

**Authors:** Sandra van Wilpe, Eveline C. F. Gerretsen, Antoine G. van der Heijden, I. Jolanda M. de Vries, Winald R. Gerritsen, Niven Mehra

**Affiliations:** 1Department of Medical Oncology, the Radboud University Medical Center, 6525 GA Nijmegen, The Netherlands; sandra.vanwilpe@radboudumc.nl (S.v.W.); Winald.Gerritsen@radboudumc.nl (W.R.G.); 2Department of Tumor Immunology, Radboud Institute for Molecular Life Sciences, Radboud University Medical Center, 6525 GA Nijmegen, The Netherlands; Eveline.gerretsen@hotmail.com (E.C.F.G.); Jolanda.deVries@radboudumc.nl (I.J.M.d.V.); 3Department of Urology, Radboud Institute for Molecular Life Sciences, Radboud University Medical Center, 6525 GA Nijmegen, The Netherlands; Toine.vanderHeijden@radboudumc.nl

**Keywords:** urothelial cancer, prognosis, biomarkers, checkpoint inhibitors, chemotherapy

## Abstract

**Simple Summary:**

There is substantial heterogeneity in the prognosis and responsiveness to registered therapies in bladder cancer. Biomarkers that can accurately predict prognosis and treatment outcome are urgently needed. Bladder cancer is considered an immunogenic tumor. In this review, we describe the available literature on the prognostic and predictive value of tumor-infiltrating immune cells and immune checkpoint expression. Several immunological markers have been associated with prognosis and treatment outcome. These markers have not yet been implemented in the clinic, likely due to the limited prognostic or predictive value of the individual markers. Future studies should, therefore, focus on combinations of biomarkers to accurately predict survival and response to treatment. The extensive overview provided here can be used to guide further biomarker research in bladder cancer.

**Abstract:**

The prognosis and responsiveness to chemotherapy and checkpoint inhibitors differs substantially among patients with bladder cancer (BC). There is an unmet need for biomarkers that can accurately predict prognosis and treatment outcome. Here, we describe the available literature on the prognostic and predictive value of tumor-infiltrating immune cells in BC. Current evidence indicates that a high density of tumor-infiltrating CD8^+^ T cells is a favorable prognostic factor, whereas PD-L1 expression and tumor-associated macrophages are unfavorable prognostic features. While PD-L1 expression appears unsuccessful as a biomarker for the response to checkpoint inhibitors, there are some indications that high CD8^+^ T cell infiltration, low transforming growth factor-beta signaling and low densities of myeloid-derived suppressor cells are associated with response. Future studies should focus on combinations of biomarkers to accurately predict survival and response to treatment.

## 1. Introduction

The immune system plays a pivotal role in the development of cancer. During malignant transformation, cells acquire tumor-associated antigens (TAAs) due to mutations in protein-encoding genes or epigenetic alterations. T cells are able to recognize these TAAs and selectively eradicate cancer cells. During progression to clinically manifest cancer, tumor cells acquire mechanisms to escape immune surveillance. Tumors may lose antigenicity through the acquisition of defects in antigen processing and the presentation or loss of immunogenic TAAs. The upregulation of inhibitory checkpoint molecules, such as programmed cell death protein-1 (PD-1) or its ligand PD-L1, provides another way to avert an effective antitumor immune response. Additionally, tumors may escape immunity by attracting immune suppressive cells, such as regulatory T cells (Tregs), tumor-associated macrophages (TAMs) and myeloid-derived suppressor cells (MDSCs). Among various cancer types, high T cell infiltration has been associated with a favorable prognosis, whereas immune suppressive cells and inhibitory immune checkpoints have been associated with a poor clinical outcome [1]. 

Bladder cancer (BC) is considered an immunogenic tumor, due to its relatively high tumor mutational burden and its responsiveness to Bacillus Calmette–Guerin (BCG) bladder instillations and checkpoint inhibitors. BC is the tenth most common malignancy worldwide, with urothelial carcinoma (UC) being by far the most common histological type of BC (>90%) [2]. Approximately 75% of patients present with non-muscle invasive bladder cancer (NMIBC). NMIBC is treated primarily by the transurethral resection of the tumor, followed by chemotherapy or BCG bladder instillations, depending on additional risk factors. Unfortunately, many patients experience disease recurrence or progression to muscle-invasive disease (MIBC) despite bladder instillations [3]. For localized MIBC, the gold standard is neoadjuvant cisplatin-based chemotherapy followed by radical cystectomy. Neoadjuvant chemotherapy induces downstaging in 44.8% of patients and pathological complete responses in 25.7%. Nevertheless, the five-year overall survival (OS) benefit is only 8%, with five-year OS increasing from 45% to 53% [4]. Recent phase II trials evaluating the role of checkpoint inhibitors in MIBC have shown that pembrolizumab and atezolizumab induce remarkable pathological complete response rates in the neoadjuvant setting (42% and 29%, resp.), suggesting that this might be an alternative treatment option for MIBC [5,6]. Long-term follow-up data, however, are still awaited. For metastatic BC (mBC), cisplatin-based chemotherapy is currently the first line of treatment. Anti-PD-(L)1 is registered for the treatment of mBC patients who progressed on platinum-based chemotherapy and patients who are cisplatin-ineligible and have a high PD-L1 expression. Although anti-PD-(L)1 is able to induce durable responses in mBC patients, objective responses and disease control (objective response or stable disease) are achieved in only 21.1% and 38.5% of patients, respectively [7].

There is a need for biomarkers that predict the prognosis and responsiveness to the available treatments in BC. Strong prognostic biomarkers could be helpful for the selection of patients who could benefit from (neo)adjuvant treatments, such as bladder instillations and chemotherapy, and to determine the intensity of the follow-up. Predictive biomarkers, on the other hand, can be used to select the most effective drugs. In this review, we will first give a comprehensive overview of the available evidence on the prognostic value of tumor-infiltrating immune cells in BC. Of note, whereas most papers discussed here focused on the immune infiltrate in UC of the bladder, some studies included a few patients with nonurothelial BC. Subsequently, we will summarize the available data on the predictive value of immunological markers for response to checkpoint inhibitors and chemotherapy. Although the majority of patients included in studies that investigated the predictive value of the immune infiltrate for response to checkpoint inhibitors were diagnosed with UC of the bladder, these studies also included patients with UC of the urethra, ureter or renal pelvis.

## 2. Prognostic Value of Tumor-Infiltrating Immune Cells 

### 2.1. CD3^+^ and CD8^+^ T Cells 

T cells are one of the key players in antitumor immunity, due to their ability to selectively kill cancer cells. There is plenty of evidence on the prognostic value of T cells in BC (Table 1) [8,9,10,11,12,13,14,15,16,17,18,19,20,21,22,23,24,25]. This is mainly derived from single-marker immunohistochemistry (IHC) studies in patients with localized BC, i.e., NMIBC and MIBC. Although some studies described an increase in T cell infiltration with a higher tumor stage and/or grade [8,14,26], most studies did not find an association with histopathological features [9,11,15,19,21,22,25,27]. When considering the prognostic value of T cells in BC, the location of the T cells appears to be important, i.e., whether they are located in the tumor epithelium, invasive margin or the surrounding stroma (Figure 1). Most studies assessing CD3^+^ or CD8^+^ T cells in the tumor epithelium or invasive margin found a trend towards or a significantly longer OS or disease-free survival (DFS) in patients with high T cell infiltration [9,16,18,19,21,22,25,26]. In contrast, out of three large studies assessing stromal T cell density, two found a negative association with survival [14,16,24].

A recent study evaluated the prognostic value of the Immunoscore in 221 BC patients undergoing a radical cystectomy [9]. The Immunoscore is a standardized IHC-based scoring system, assessing the CD3^+^ and CD8^+^ T cell densities in the tumor epithelium and the invasive margin. It has been validated in colorectal cancer and tested in several other tumor types [28]. The Immunoscore was found to be an independent predictor of OS in BC [9], with OS being significantly longer in patients with a high Immunoscore. Its prognostic value was stronger than the tumor or nodal status or the presence of vascular invasion (HR 2.01 (low vs. high); *p* = 0.008) [9]. A smaller study, including 67 BC patients, found a significant association with DFS (HR 0.13; *p* = 0.02), but not OS [18].

Apart from the Immunoscore, tumors can be also classified into three immune phenotypes, based on the presence of CD8^+^ T cells in the intraepithelial and stromal compartment, i.e., immune-desert, inflamed and immune-excluded tumors (Figure 1). In immune-desert tumors, there are hardly any T cells present in the intraepithelial or stromal compartment. Inflamed tumors, on the other hand, are highly infiltrated by T cells, with T cells present in both compartments. In immune-excluded tumors, T cells can be found in the stroma, but they are unable to penetrate the tumor epithelium. In MIBC, the immune-desert phenotype appears to be most common (63%), with only 21% and 16% of patients having an immune-excluded and inflamed phenotype, respectively [23]. In mBC, the immune-excluded phenotype is more common (47%), and immune-desert and inflamed phenotypes are seen in 27% and 26% of patients, respectively [29]. A study in 258 MIBC patients demonstrated significant survival differences between the three phenotypes, with the five-year OS rates being 46.6%, 70.1% and 79.7% (*p* < 0.001) in patients with an immune-desert, immune-excluded and inflamed phenotype [23]. The classification of tumors into these immune phenotypes could provide an easy prognostic tool in BC.

Whereas most studies in BC used IHC to evaluate immune cell infiltration, it is also possible to infer the immune cell composition from bulk RNA-sequencing data (see Box 1). In BC, three studies used RNA sequencing to study the prognostic value of T cell infiltration. The studies used different methods, but had (partially) overlapping study populations, with data being derived from (a subset of) BC patients included in The Cancer Genome Atlas (TCGA) [12,13,24]. One study evaluated CD3^+^ T cell infiltration and described a positive correlation with OS, with median OS being 819 days in patients with low CD3^+^ T cell infiltration and 2828 days in patients with high CD3^+^ T cell infiltration [13]. RNA-sequencing studies did not find a significant correlation between CD8^+^ T cell infiltration and the clinical outcome. Considering the importance of T cell location, this is not unexpected, as it is impossible to locate immune cells in intraepithelial or stromal regions when using bulk RNA sequencing. 

In summary, T cell infiltration in the tumor epithelium or invasive margin is a favorable prognostic factor in BC and is best evaluated by IHC. The prognostic values of the Immunoscore or immune phenotype in BC need further validation, but are promising prognostic tools in BC.

### 2.2. B Cells

So far, research on tumor-infiltrating immune cells in BC has mostly focused on T cells and immune suppressive cells. However, the density of infiltrating B cells might also have a prognostic utility. Li and colleagues [12] used the RNA-sequencing data of 404 MIBC patients included in the TCGA to study the prognostic value of B cells. In a multivariate analysis, including CD8^+^ and CD4^+^ T cells, neutrophils, macrophages, DCs, age and tumor stage, B cells were an independent favorable prognostic factor (HR 0.02, *p* = 0.022). Median OS for the 20% of patients with the highest B cell infiltration was 2000 days, whereas patients with the lowest B cell infiltration (bottom 20%) had a median OS of only 575 days. Fu and colleagues [24] used the same RNA-sequencing data, but differentiated between plasma cells, memory B cells and naïve B cells. There was a trend towards more favorable OS for memory B cells and plasma cells, without reaching statistical significance. IHC studies on the prognostic value of B cells in BC are limited. One IHC study in 115 NMIBC patients found no difference in the amount of tumor-infiltrating B cells between patients with and without recurrent disease [10]. Further research is needed to find out whether B cells might be useful as a prognostic marker in BC.

### 2.3. Dendritic Cells 

Dendritic cells (DCs) are important players in antigen-specific immunity. They capture antigens and present them to T cells via major histocompatibility complex (MHC) molecules. Antigen presentation by DCs can lead either to an immune response or to immune tolerance, depending on their maturation and concomitant immunomodulatory signals. Data on the prognostic value of DCs in BC is sparse. An IHC study in 93 NMIBC patients observed that patients with high numbers of tumor-infiltrating DCs, defined as CD83^+^ cells, more frequently progressed to MIBC in the absence of BCG instillations when compared to patients with low numbers of DCs (HR 8.25, *p* = 0.018) [30]. CD83^+^ cells appeared to be particularly useful as a predictive marker in patients aged 70 years or older, with the six-year progression-free survival (PFS) being at 87% in patients with few CD83^+^ cells and at 56% in patients with high levels of CD83^+^ cells. In another study including 30 NMIBC patients, high levels of CD83^+^ DC were associated with poor recurrence-free survival (RFS) following BCG instillations (HR 9.81, *p* = 0.039) [31]. Although CD83 is expressed on the surface of activated DCs and causes the upregulation of markers that are required for T cell activation, several studies have also suggested an immunoregulatory role for CD83 [32], possibly explaining the negative association with the clinical outcome in these IHC studies. RNA-sequencing studies have yielded conflicting results regarding the prognostic value of DCs in BC [12,24]. It has been reported that current computational methods perform poorly in estimating the abundance of DCs from RNA-sequencing data [33].

### 2.4. Natural Killer Cells

CD8^+^ T cells require the presentation of antigens in the context of MHC-I molecules. The downregulation of MHC-I appears to be an important mechanism by which tumors evade the immune system [5]. Under conditions of a low MHC-I expression, natural killer (NK) cells, which mediate cytotoxicity in an MHC-unrestricted fashion, might play an important role in antitumor immunity. It has been described that NK cells are relatively abundant in BC when compared to other immune cells, accounting for approximately 20% of tumor-infiltrating leukocytes [34]. However, an IHC study in 115 NMIBC patients found no association between CD56^+^ NK cells and disease recurrence [10]. Similarly, the density of intraepithelial CD56^+^ NK cells did not correlate with clinical outcome in MIBC (*n* = 258) [24]. Although there does not appear to be an association between the total NK cell population and survival, there are some indications that specific subgroups of NK cells have a prognostic value. Mukherjee and colleagues [34] differentiated between CD56^bright^ and CD56^dim^ NK cells in a small cohort consisting of 20 NMIBC and 30 MIBC patients. A positive association with survival was found for CD56^bright^, but not CD56^dim^, NK cells. CD56^bright^ NK cells are considered an immunoregulatory subset of NK cells [35], which exhibit an enhanced IFN-γ production and lower cytotoxicity than their CD56^dim^ counterparts [34]. In addition, an RNA-sequencing study differentiated between activated and resting NK cells and found a positive correlation between activated NK cells and OS in MIBC (*n* = 287, HR 0.0001, *p* = 0.018) [24]. These findings require further evaluation and validation.

### 2.5. Tregs

Tregs function as suppressors of antitumor immunity. Tregs can be identified by the expression of the transcription factor forkhead box protein P3 (FoxP3). A meta-analysis showed that Treg density is associated with shorter OS in the majority of solid tumors [36]. Studies on the prognostic value of Tregs in BC are limited and have yielded inconsistent results [8,15,20,24]. Interestingly, Siefker and colleagues [15] studied the ratio between Tregs and CD3^+^ T cells. They included 149 patients, most of whom had MIBC. Although there was a trend towards poorer OS with increasing Treg infiltration (*p* = 0.17), only the FoxP3/CD3 ratio was significantly associated with OS (HR 1.29, *p* = 0.016), with five-year OS being at 58% and 20% in patients with a low and high FoxP3/CD3 ratio, respectively. It is possible that not the density of Tregs, but rather the number of Tregs in relation to the total number of T cells, is prognostic in BC. 

### 2.6. TAMs

Macrophages are versatile cells. They are often classified into proinflammatory (M1) and anti-inflammatory (M2) macrophages. Previous studies have shown that TAMs often have an M2 phenotype and promote tumor progression and metastasis by mechanisms such as angiogenesis and matrix breakdown [37]. In IHC studies, CD68 is often used to identify TAMs. Additional markers can be used to differentiate between M1 (CD169) or M2 (CD163 or CD204) macrophages [38,39]. 

Several studies have identified a higher density of (M2) macrophages in BC patients with unfavorable histopathological features, including a high tumor and nodal stage and histological grade [8,40,41]. Almost all studies, including both IHC and RNA-sequencing studies, described a trend towards or a significantly shorter survival in patients with high densities of (M2) macrophages [8,10,12,24,30,40,42,43,44,45] (Table 2). A comprehensive and accurate evaluation of TAMs in BC has been performed by Wang and colleagues [40]. In this study, both the intraepithelial and stromal density of total, M1 and M2 TAMs was evaluated in 302 patients with localized BC. The intraepithelial and stromal density of CD68^+^, CD169^+^ and CD204^+^ cells all correlated negatively with OS in the univariate analysis, except for the intraepithelial CD68^+^ cell density. In the multivariate analysis, however, only the association with stromal CD204^+^ macrophages was statistically significant (HR 1.98, 95% CI 1.10–3.56, *p* = 0.022). The five-year OS was at 70% in patients with stromal CD204^+^ cell counts above the median, and at 91% for patients with cell counts below the median. Interestingly, another IHC study described that, although CD68^+^ was not associated with cancer-specific survival (CSS), a high CD68/CD3 ratio was a strong, independent predictor of poor CSS (HR 7.73, *p* = 9.5 × 10^−6^). Three-year CSS was only at 18% in patients with a CD68/CD3 ratio above 1, compared to being at 70% in patients with a ratio below 1 [8]. 

### 2.7. MDSCs

MDSCs comprise a heterogeneous population of immature immune suppressive myeloid cells, which can be classified into monocytic (M-MDSCs), polymorphonuclear (PMN-MDSCs) and early MDSCs (eMDSCs). MDSCs protect tumors from elimination by CD8^+^ T cells and promote tumor growth. Zhang and colleagues [46] studied tumor-infiltrating MDSCs in 116 newly diagnosed BC patients. The number of MDSCs, defined as CD33^+^ cells, was significantly increased in tumor tissue compared to tumor-adjacent tissues. In a cohort of 85 BC patients undergoing a cystectomy, eMDSCs (CD33^+^HLA-DR^-^CD15^-^CD14^-^) were the predominant subtype of MDSCs in 57% of patients and PMN-MDSCs (CD33^+^HLA-DR^-^CD15^+^CD14^-^) in 43%. The number of M-MDSCs (CD33^+^HLA-DR^-^CD15^-^CD14^+^) was low compared to other MDSC subsets [47]. Data on the prognostic value of MDSCs in BC is sparse. Zhang and colleagues [46] described that high levels of tumor-infiltrating CD33^+^ cells were associated with an advanced disease stage and poor OS (*p* < 0.01, *n* = 116). In accordance with this, high numbers of MDSCs in blood and urine have also been associated with poor OS in BC [48,49], suggesting that MDSC might have a prognostic utility in BC.

### 2.8. Neutrophils

Neutrophils are the most abundant circulating immune cells. They represent the first line of immune defense against invading pathogens. A meta-analysis, including 20 studies in 10 different solid tumors, described that a high density of tumor-infiltrating neutrophils was an independent predictor of unfavorable DFS and OS [50]. Liu and colleagues [11] studied the prognostic value of neutrophils in 102 patients with localized BC. High tumor-infiltrating neutrophils were associated with a high tumor stage and histological grade and were an independent predictor of poor OS (HR 2.43, *p* = 0.044). The relation between the neutrophil-to-lymphocyte ratio (NLR) and OS was even stronger (HR 3.53, *p* = 0.028) [11]. Zhou and colleagues [51] studied the prognostic value of neutrophils in two IHC cohorts (*n* = 142 and *n* = 119). In both cohorts, patients with high tumor-infiltrating neutrophils showed significantly poorer OS when compared to patients with low tumor-infiltrating neutrophils. When adjusting for the pathological stage, histological grade, age and whether or not adjuvant chemotherapy was given, HRs for OS were 2.12 (*p* = 0.007) and 3.81 (*p* < 0.001) [51]. Although the results of these IHC studies are promising, it is important to note that the studies used CD66b or CD15 expression as a neutrophil marker and did not differentiate between true neutrophils or PMN-MDSCs. RNA-sequencing studies have yielded inconsistent results regarding the prognostic value of neutrophils [12,24,51]. 

### 2.9. Eosinophils

Eosinophils are a subset of granulocytes mostly known for their role in parasite infections and allergies. Eosinophils have been associated with a favorable prognosis in several cancer types, but a poor prognosis in a few others [52]. Little is known about the prognostic role of eosinophils in BC. Retrospective analyses on RNA-sequencing data of the TCGA database did not reveal a correlation between eosinophils and overall survival in MIBC [24,52]. 

### 2.10. Mast Cells

Tumor infiltration by mast cells, which are well known for their role in allergies and anaphylaxis, has been described in BC. Preclinical studies in BC indicate that mast cells contribute to angiogenesis [53] and promote tumor progression and metastasis [54]. Fu and colleagues assessed the prognostic value of mast cells in three cohorts of MIBC patients. In an RNA-sequencing cohort consisting of 287 patients, there was a nonsignificant negative association between the mast cell density and OS (HR 96.57, 95% CI 0.55–16,929.29, *p* = 0.08). In one of the two IHC cohorts, there was a significant negative association between the stromal mast cell density and OS (cohort 1: *n* = 118, HR 1.04, 95% CI 1.00–1.09, *p* = 0.07; cohort 2: *n* = 140, HR 1.03, 95% CI 1.01–1.06, *p* = 0.001) [24].

### 2.11. Immune Checkpoint Molecules

Checkpoint molecules play an important role in the regulation of immune responses. In BC, there is plenty of data on the prognostic value of PD-L1 expression. A recent meta-analysis, including 11 studies totaling 1697 BC patients, described that a high PD-L1 expression on tumor cells was associated with a high tumor stage (T2-4 vs. Ta-1; OR 3.9, *p* < 0.001) and the presence of distant metastasis (OR 2.5, *p* = 0.012). High PD-L1 expression was also associated with poor OS (HR 1.83, *p* = 0.002). There was a nonsignificant association between the PD-L1 expression and CSS (HR 1.51, *p* = 0.20), RFS (HR 1.43, *p* = 0.13) and DFS (HR 1.53, *p* = 0.13) [55]. Importantly, all of the included studies were conducted before the registration of checkpoint inhibitors for the treatment of BC. Interestingly, two large studies in the meta-analyses did not find a significant correlation between PD-L1 expression and OS in the total study population, but only in the subset of patients with organ-confined disease [56,57]. Whereas the five-year OS difference in patients with organ-confined disease was limited in one study (63% versus 69%, *n* = 302) [56], a larger difference was observed by Boorjian and colleagues (41.2% versus 68.6%, *n* = 318) [57]. While all studies included in the meta-analysis evaluated PD-L1 expression on tumor cells, only three studies evaluated PD-L1 expression on immune cells, with contradictory results [14,58,59].

Data on the prognostic value of other checkpoint molecules in BC is limited. Yang and colleagues [60] studied the prognostic value of PD-1 and T-cell immunoglobulin mucin-3 (TIM-3) in the tumor tissue of 100 patients with localized BC. They described that TIM-3 was aberrantly expressed in 96% of patients. PD-1 expression was observed in 82% of patients and significantly correlated with TIM-3 expression (*r* = 0.341, *p* = 0.001). TIM-3 and PD-1 expression were both indicators of poor DFS (TIM-3: RR 5.4, *p* = 0.011; PD-1: RR 4.2, *p* = 0.003) and OS (TIM-3: RR 5.5, *p* = 0.009; PD-1: RR 3.3, *p* = 0.010). The expression of both markers was associated with a significantly worse prognosis when compared to either of the individual markers (*p* = 0.002 for DFS, *p* < 0.001 of OS). In contrast, two large studies (*n* = 314 and *n* = 302) found no significant association between PD-1 expression and survival [56,57]. 

### 2.12. Summary

An overview of the prognostic value of the various immune cell types is given in Figure 2 (IHC) and Figure 3 (RNA sequencing). In summary, there is plenty of evidence that a high CD8^+^ T cell density in the tumor epithelium or invasive margin, low TAM infiltration and low PD-L1 expression on tumor cells are favorable prognostic factors in localized BC. Nevertheless, these markers have not yet been implemented in the clinic, likely due to the limited prognostic value of the individual markers. As described, a few small studies have investigated the prognostic value of combinations of markers, i.e., FoxP3/CD3 ratio, CD68/CD3 ratio and NLR, and found that the combination had a stronger prognostic value compared to either marker alone. Further research on prognostic markers in BC should focus on combinations of biomarkers. These combinations need not necessarily be restricted to the immune markers discussed here, but could also include other parameters that have been shown to have a prognostic value in BC, such as the molecular subtype [61]. 

Box 1Background information on immunohistochemistry and RNA sequencing.
Immunohistochemistry: A common method to quantify tumor-infiltrating immune cells is immunohistochemistry (IHC). Most studies included in this review used single-marker IHC. An advantage of IHC is the ability to study immune cells in their spatial context, which makes it possible to distinguish between immune cells located in the tumor epithelium, invasive margin or surrounding stroma. A disadvantage of single-marker IHC is that it utilizes only one marker per test, whereas, for the phenotypic characterization of some cell types (i.e., MDSCs), multiple markers are needed. However, recent advances in multiplex immunohistochemistry and multispectral imaging now enable the simultaneous analysis of multiple tissue markers. Another disadvantage of single-marker IHC is that it is laborious and has a low throughput. Although advances are made in the automated analysis of IHC images, stainings are still often visually assessed by pathologists. Most studies included in this review used either 1.0-mm tissue microarrays (TMAs) or selected a limited number of fields from whole slides for analyses (mostly 0.07 mm^2^/field). It is questionable whether these small regions accurately reflect the tumor immune infiltrate. A recent study in NMIBC reported that two to six 0.6-mm TMAs are needed to provide a correct sampling of NMIBC tumors because of spatial heterogeneity [26].RNA sequencing: Various computational tools have been developed to estimate the relative abundance of immune cells in the tumor using bulk RNA data. Studies included in this review have used CIBERSORT, TIMER or measured CD8A and CD3D gene expression. Where CIBERSORT determines the fraction of 22 immune cell types relative to the total immune cell content, TIMER measures the abundance of six immune cell types. Currently available computational tools have shown good correlations for CD8^+^ T cells, B cell, NK cells and macrophages. However, nonregulatory and regulatory CD4^+^ cells are hard to distinguish. Moreover, DCs cannot be accurately quantified, and none of the current methods take MDSCs into account [33]. A disadvantage of the use of bulk RNA data for immune cell quantification is that it is not possible to study the location of the immune cells.


## 3. The Predictive Value of Tumor-Infiltrating Immune Cells for Response to Therapy

Besides the need for prognostic markers, there is also an urgent need for biomarkers that can be used to identify patients that benefit from specific treatments, such as chemotherapy and immune checkpoint inhibitors.

### 3.1. Immune Checkpoint Inhibitors

Since 2017, immune checkpoint inhibitors are used for the treatment of mBC. The use of checkpoint inhibitors in BC is expanding. In January 2020, pembrolizumab was approved by the FDA for the treatment of BCG-unresponsive, high-risk NMIBC, based on the results of the KEYNOTE-057 [62]. Additionally, phase II clinical trials evaluating the efficacy of neoadjuvant checkpoint inhibitor therapy in localized MIBC have shown promising results [5,6]. Unfortunately, not all patients benefit from checkpoint inhibitors. In mBC, objective responses and disease control are seen in only 21.1% and 38.5% of patients, respectively [7]. There is a need for predictive biomarkers that can be used to identify these patients. 

Most evidence on biomarkers for checkpoint inhibitor response prediction in BC came from clinical trials that included patients with UC of the bladder, urethra, ureter and renal pelvis. The biomarker that has been most extensively studied for its predictive value is PD-L1. Several tests have been developed to assess PD-L1 expression. Each therapeutic antibody has its own companion diagnostic, with a corresponding cut-off value, and some variation can be observed between assays [63]. A recent meta-analysis, including eight studies totaling 1436 patients, evaluated the use of PD-L1 as a biomarker for the response to PD-(L)1 inhibitors in advanced UC. Patients had received pembrolizumab, nivolumab, atezolizumab, durvalumab or avelumab. Patients with high PD-L1 expression had significantly higher objective response rates compared to patients with low PD-L1 expression (RR 0.53, *p* < 1 × 10^6^). However, it did not accurately predict one-year OS (RR 0.53, *p* = 0.304) [64]. To determine whether a biomarker is truly predictive of response rather than prognostic, a control arm is needed, consisting of patients that did not receive the treatment of interest. In UC, there are two available studies that assessed PD-L1 expression and included a control arm. The IMvigor211 [65] is a randomized controlled, phase III trial comparing the PD-L1 inhibitor atezolizumab with second-line chemotherapy in metastatic UC (mUC). Based on the hypothesis that the efficacy of atezolizumab would be associated with PD-L1 expression, the investigators used a hierarchical study design where OS was first evaluated in the group of patients with PD-L1 expression on at least 5% of tumor-infiltrating immune cells (IC2/3). A significant difference would have allowed for further analysis in the intention-to-treat population, also including patients with lower PD-L1 expression. Atezolizumab, however, did not prolong OS in the IC2/3 population. High PD-L1 expression was associated with higher response rates in both the atezolizumab and chemotherapy arms. The other study including a control arm is the KEYNOTE-045 [66], a randomized controlled, phase III trial comparing pembrolizumab with second-line chemotherapy. In the KEYNOTE-045, the benefit of pembrolizumab appeared to be independent of PD-L1 expression on tumor and immune cells. These studies indicate that PD-L1 expression is not useful as a predictive biomarker in mUC. However, in 2018, the use of pembrolizumab and atezolizumab as a first-line treatment in cisplatin-ineligible mUC patients was restricted to patients with high PD-L1 expression, based on unpublished, interim analyses of the KEYNOTE-361 (NCT02853305) and IMvigor130 (NCT02807636) performed by the data safety monitoring committees. Both trials compared the combination of checkpoint inhibitors and chemotherapy with checkpoint inhibitors or chemotherapy alone in treatment-naïve mUC patients and observed poor survival in patients that received first-line single-agent checkpoint inhibitors. With both studies still ongoing, it is uncertain whether PD-L1 expression is truly a predictive biomarker in this setting. It is also possible that anti-PD-(L)1 is inferior to chemotherapy in mUC, regardless of the PD-L1 expression in the first-line setting. 

In addition to PD-L1 expression, several studies have evaluated the predictive value of CD8^+^ T cells for the response to anti-PD-(L)1. In 212 mUC patients receiving nivolumab as a second-line treatment, a high CD8^+^ T cell infiltration was associated with a favorable clinical outcome. Objective responses were observed in 25.5% and 11.3% of patients with high and low CD8^+^ T cell infiltration, respectively. The median OS was 11.3 and 5.72 months [67]. Similarly, in the IMvigor 210, a phase II trial evaluating the efficacy of atezolizumab in mUC, CD8^+^ T cell infiltration was higher in patients who responded to atezolizumab (*p* = 0.0265) [68]. Moreover, phase I studies evaluating the efficacy of atezolizumab have also described a positive association between effector T cell signatures and the clinical outcome [69,70]. In line with the findings in mUC, the results from the ABACUS trial, a phase II trial investigating the efficacy of neoadjuvant atezolizumab in UC of the bladder, indicate that CD8^+^ T cell infiltration is also associated with higher pathologic complete response rates to neoadjuvant atezolizumab (40% vs. 20% (high vs. low), *p* < 0.05). Moreover, a predefined eight-gene cytotoxic T cell signature was significantly increased in responders when compared to patients with stable disease (*p* < 0.01) and patients who relapsed (*p* < 0.01) [71]. 

A lack of response to atezolizumab in mUC has been associated with a signature of transforming growth factor-β (TGF-β), which was found to be particularly high in patients with immune-excluded tumors. Interestingly, in preclinical models, the combination of anti-TGF-β and anti-PD-L1 was shown to reduce TGF-β signaling, increase intraepithelial T cell penetration and significantly reduce the tumor size in mice with high TGF-β signaling and immune-excluded tumors [29]. 

Finally, a phase I trial evaluating the efficacy of atezolizumab in 68 UC patients found that patients with high baseline signatures of genes associated with myeloid cells (IL-1B, IL-8) had lower response rates to atezolizumab (*p* < 0.01) [70]. In line with these results, an exploratory biomarker analysis in the Checkmate 275 trial, a single-arm phase II trial evaluating the efficacy of nivolumab after first-line chemotherapy in 220 mUC patients, suggests that low baseline circulating MDSC levels are indicative of a longer OS after nivolumab treatment (*p* < 0.05) [72]. 

To summarize, although PD-L1 expression has been associated with higher response rates to checkpoint inhibitors, it does not appear to be useful as a predictive biomarker in mUC. Studies have suggested that CD8^+^ T cell infiltration, TGF-β signaling and MDSCs are associated with the response to checkpoint inhibitors. However, data are limited, and, due to the lack of a control arm, it is not possible to determine whether these biomarkers are truly predictive or merely prognostic. 

In light of the presumed relationship between the immune infiltrate and response to checkpoint inhibitors, the timing of checkpoint inhibitor initiation might be crucial. Recently, a randomized, phase III trial demonstrated that maintenance therapy with the PD-L1 inhibitor avelumab, following response or stable disease with first-line chemotherapy, significantly prolonged OS. The median OS with avelumab, plus best supportive care, was 21.4 months compared to 14.3 months with best supportive care alone (HR 0.69; 95% CI 0.56–0.86) [73]. The success of this treatment strategy might be related to the immune modulatory effect of the chemotherapeutic agents. Although chemotherapy has historically been thought to act through the direct killing of tumor cells, accumulating evidence indicates that chemotherapeutics, including cisplatin and gemcitabine, also have immunomodulatory effects. Preclinical evidence indicates that cisplatin increases the infiltration and cytotoxic activity of CD8^+^ T cells [74,75,76]. In addition, cisplatin and gemcitabine have also been described as increasing PD-L1 expression [77,78] and reducing the abundance of MDSCs [75,79,80,81], Treg [82,83] and macrophages [84]. Limited studies have evaluated the effects of cisplatin and gemcitabine on antitumor immunity in UC patients. In a cohort of 20 MIBC patients, NK cells significantly increased and M2 macrophages significantly decreased after neoadjuvant chemotherapy (MVAC) [24]. Another study including 40 MIBC patients demonstrated that the total T cell density was significantly higher in the sentinel nodes of patients that had received chemotherapy compared to chemotherapy-naïve patients, whereas the frequency of Tregs was lower [85]. These changes, occurring during platinum-based chemotherapy, might make patients more susceptible to subsequent treatment with PD-(L)1 inhibitors.

### 3.2. Chemotherapy 

As discussed above, accumulating evidence indicates that platinum-based chemotherapy exerts some of its effects via immune modulation. It is important to understand how the immune infiltrate influences the response to both checkpoint inhibitors and chemotherapy, in order to make well-informed treatment decisions. Few studies have evaluated the predictive value of tumor-infiltrating immune cells for the response to chemotherapy in BC. One study, including 41 patients, studied the relationship between the CD8^/^FoxP3 ratio and response to neoadjuvant chemotherapy. Strikingly, the response rates to neoadjuvant chemotherapy were 60%, 42% and 0% for tumors with high, intermediate and low CD8/FoxP3^+^ ratios, respectively. No association was observed between PD-L1 expression and response [86]. In another study, including 22 MIBC patients treated with neoadjuvant chemotherapy, neither CD8^+^ T cell infiltration nor PD-L1 expression was associated with the response to chemotherapy [87]. 

Recently, researchers aimed to construct an immunotype to predict survival and the benefit of adjuvant chemotherapy in MIBC patients. In total, 545 patients were classified into stromal immunotype A (CTL^high^NK^high^Treg^low^Macrophage^low^MastCell^low^) or stromal immunotype B (CTL^low^NK^low^Treg^high^Macrophage^high^MastCell^high^). Compared to immunotype A, immunotype B had poor five-year OS (76.0% vs. 44.0%, *p* < 0.001) and RFS (62.8% vs. 48.3%, *p* < 0.001). Adjuvant chemotherapy did not enhance OS or RFS in either the pathological (p)T2 (tumors infiltrating the detrusor muscle but not the perivesical tissue) or pT3-4 subgroup (tumors invading the perivesical tissue). After grouping the patients into immunotypes A and B, adjuvant chemotherapy significantly enhanced OS and RFS in patients with immunotype B and pT3-4 disease (HR for OS 0.60, *p* = 0.016; HR for DFS 0.56, *p* = 0.006) [24]. What is seemingly conflicting, however, is that chemotherapy also seemed to prolong OS and DFS in patients with pT2 disease and immunotype A. The same research group reported on the predictive value of mast cells in a separate paper [88]. In patients with a low stromal mast cell infiltration, adjuvant chemotherapy prolonged OS and RFS (*p* = 0.032 and *p* = 0.011, resp.), whereas this was not the case for patients with a high stromal mast cell density.

Lastly, Qi and colleagues [43] studied whether galectin-9^+^ TAMs could be used to predict benefits from adjuvant chemotherapy in 75 MIBC patients. Galectin-9 is thought to promote the differentiation of macrophages towards an M2 phenotype [89]. Whereas the total population did not benefit from adjuvant chemotherapy (*p* = 0.305), chemotherapy significantly prolonged OS and RFS in patients with high galectin-9^+^ TAMs (HR for OS 0.22, *p* < 0.001; HR for RFS 0.15, *p* = 0.001). 

In conclusion, small studies have indicated that the CD8/FoxP3 ratio, the density of mast cells and the density of galectin-9^+^ TAMs might be predictive of the response to chemotherapy. More research is needed to elucidate whether immune markers can be used in the clinic to predict the response to chemotherapy in BC. 

## 4. Conclusions

There is a need for prognostic biomarkers in BC that can be used to select patients for adjuvant treatments, such as bladder instillations and (neo)adjuvant chemotherapy. The current literature demonstrates that intraepithelial, but not stromal, CD8^+^ T cell infiltration is a favorable prognostic factor in localized BC. TAMs and PD-L1 expression, on the other hand, are unfavorable features. None of these markers have been implemented in the clinic, possibly due to the limited prognostic value of individual markers. Further research is needed to study whether combinations of these markers, possibly combined with clinical, histological or molecular biomarkers, might provide a strong prognostic value and could be used in the clinic to guide treatment decisions. 

So far, no biomarker has been identified that can accurately predict the response to immune checkpoint inhibitors in BC. However, high PD-L1 expression, high CD8^+^ T cell infiltration, low TGF-β signaling and low MDSCs have been associated with higher response rates to checkpoint inhibitors. In this regard, it is of interest that cisplatin-based combination chemotherapy appears to increase CD8^+^ T cell infiltration and PD-L1 expression and decreases the number of immune suppressive cells. Further research is warranted to elucidate the predictive value of tumor-infiltrating immune cells for the response to chemotherapy and checkpoint inhibitors in BC.

## Figures and Tables

**Figure 1 cancers-12-02692-f001:**
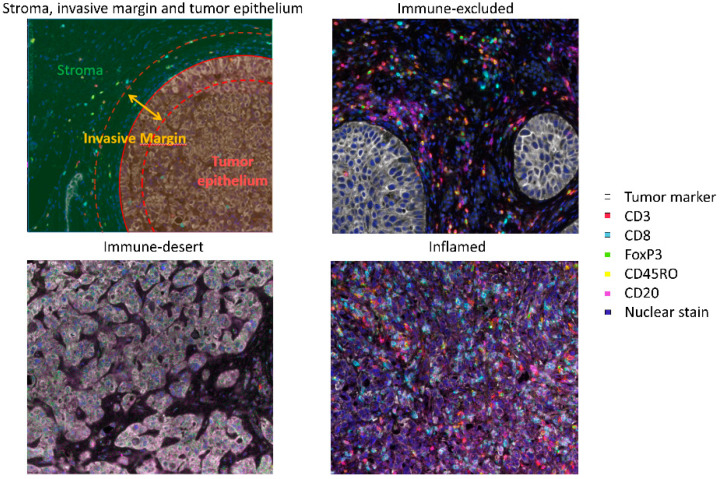
The upper left panel illustrates how tumor tissue can be segmented into the stroma, invasive margin and tumor epithelium. The invasive margin, sometimes also called the peritumoral region, is defined as the border of the tumor nest, including a small stromal and a small intraepithelial region. In the other panel, multiplex immunohistochemistry images of bladder cancer specimens with an immune-desert (lower left), immune-excluded (upper right) and inflamed (lower right) phenotype are shown.

**Figure 2 cancers-12-02692-f002:**
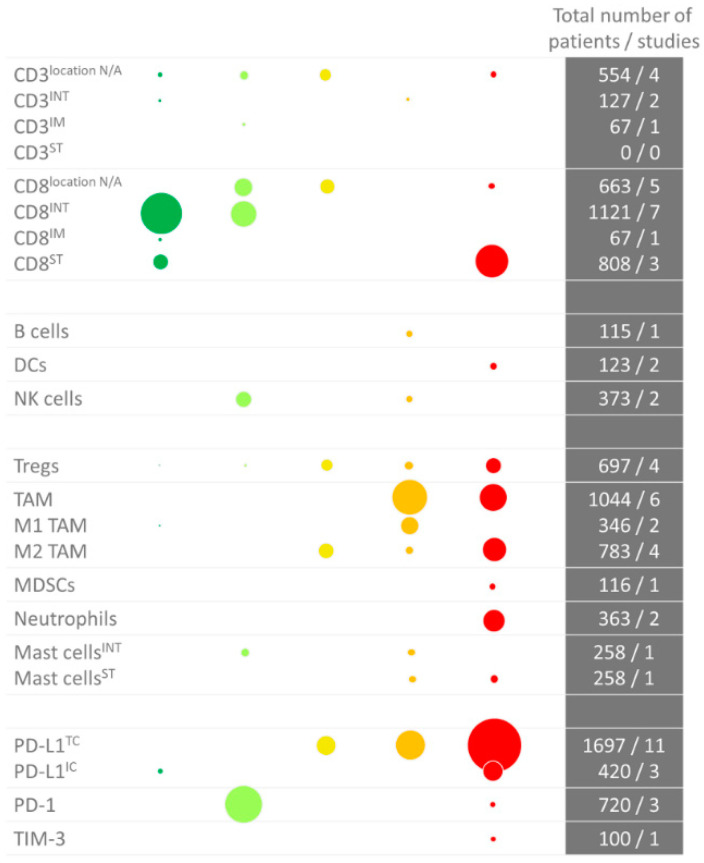
Overview of immunohistochemistry studies in bladder cancer per immune cell type. The color (and position) indicate whether there was a positive or negative correlation. Dark green: significant positive correlation; light green: trend towards a positive correlation; yellow: no trend; orange: trend towards a negative correlation; red: negative correlation. The size of the dots indicates the number of studied patients. As some studies performed an analysis in more than one cohort, the number of dots is sometimes higher than the total number of studies (i.e., for the Tregs). Abbreviations: DCs: dendritic cells; IC: immune cell; INT: intraepithelial; IM: invasive margin; MDSCs: myeloid-derived mononuclear cells; N/A: not applicable; NK cells: natural killer cells; PD-(L)1: programmed death (ligand)-1; ST: stroma; TAM: tumor-associated macrophage; TC: tumor cell; TIM-3: T-cell immunoglobulin mucin-3; Tregs: regulatory T cells.

**Figure 3 cancers-12-02692-f003:**
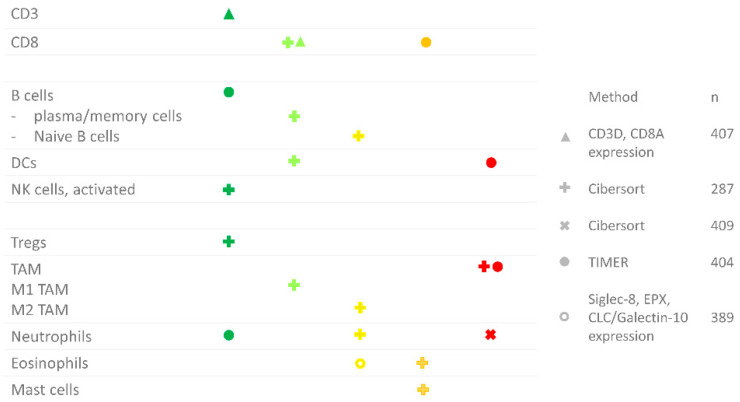
Overview of RNA sequencing studies in bladder cancer per immune cell type. The color (and position) indicate whether there was a positive or negative correlation. Dark green: significant positive correlation; light green: trend towards a positive correlation; yellow: no trend; orange: trend towards a negative correlation; red: negative correlation. The shape indicates the study where the data was derived from: **∆** [13], **+** [24], **x** [51], **●** [12], **o** [52].

**Table 1 cancers-12-02692-t001:** Prognostic value of infiltrating CD8^+^ and CD3^+^ T cells.

Ref	*N*	Stage	UC only	Methods	Relation with Survival					
					Outcome Parameter	Immune Marker	HR ^1^	95% CI	*p*	Adjusted for
[10]	115	Low-grade NMIBC	Yes	IHC	RFS	CD3	5.83 ^2^	1.52–22.36	0.010	Tumor size, T stage, GZMB and CD4,20,56,68
						CD8	5.33 ^2^	1.51–18.74	0.009	
[25]	302	NMIBC: 212 MIBC: 90	Yes	IHC	OS	CD8 ^INT^	0.87	0.46–1.65	0.67	Age, T and N stages, tumor size and grade
					RFS	CD8 ^INT^	0.68	0.46–1.01	0.057	-
[21]	69	NMIBC: 38 MIBC: 31	Yes	IHC	OS	CD8 ^INT^	0.3	0.09–0.96	0.042	T stage
					DFS	CD8 ^INT^	0.79	0.27–2.33	0.67	
[8]	253	NMBIC: 201 MIBC: 52	Yes	IHC	PFS - NMIBC	CD3, CD8	NR	NR	ns	-
					CSS - MIBC	CD3	0.60	0.41–0.89	0.009	
						CD8	0.82	0.57–1.16	0.26	
[18]	67	NMIBC: 5 MIBC: 62	No	IHC	OS	CD3 ^INT^	2.91	0.85–10.0	0.09	Stage, LVI, surgical margins, prior BCG and (N)AC
						CD3 ^IM^	0.57	0.12–2.63	0.47	
						CD8 ^INT^	0.32	0.09–1.11	0.07	
						CD8 ^IM^	0.33	0.12–0.88	0.03	
					DFS	CD3 ^INT^	1.27	0.46–3.53	0.64	
						CD3 ^IM^	0.60	0.16–2.17	0.43	
						CD8 ^INT^	0.55	0.18–1.65	0.29	
						CD8 ^IM^	0.35	0.14–0.86	0.02	
[11]	102	NMIBC: 53 MIBC: 49	Yes	IHC	OS	CD8	0.66	0.28–1.58	0.35	T stage, tumor size, grade, NLR and neutrophils
[19]	60	T1	Yes	IHC	PFS	CD3 ^INT^	NR	NR	0.69	-
					CSS	CD3 ^INT^	NR ^3^	NR	0.045	
[20]	37	NMIBC: 4 MIBC: 33	Yes	IHC	OS	CD3 ^INT/IM^	0.42	0.18–1.00	0.049	Age, gender, T and M stages, and (N)AC
					PFS	CD3 ^INT/IM^	0.09	0.02–0.48	0.004	
[22]	56	NMIBC: 11 MIBC: 45	Yes	IHC	OS	CD8 ^INT^	0.1	0.01–0.69	0.02	T and N stages, and LVI
					CSS	CD8 ^INT^	0.05	0.01–0.62	0.02	
[12]	404	MIBC ^4^	Yes	RNA-seq, TIMER	OS	CD8	7.70	1.60–36.97	0.12	Age, stage, B cells, DCs, neutrophils and TAMs
[24]	287	MIBC ^4^	Yes	RNA-seq, CIBERSORT	OS	CD8	0.15	0.014–1.75	0.13	-
	118	MIBC	No	IHC	OS	CD8 ^INT^	0.91	0.84–0.98	0.016	
						CD8 ^ST^	0.97	0.94–0.99	0.003	
	140	MIBC	No	IHC	OS	CD8 ^INT^	0.94	0.90–0.99	0.015	
						CD8 ^ST^	0.98	0.97–0.99	0.001	
[13]	407	MIBC ^4^	Yes	RNA-seq	OS	CD3D	NR ^5^	NR	0.032	Age, gender and T stage
						CD8A	NR ^5^	NR	0.06	
[14]	248	NMIBC: 129 MIBC: 119	Yes	IHC	OS	CD8 ^ST^	NR ^6^	NR	<0.01	-
					RFS	CD8 ^ST^	NR	NR	0.99	
[15]	149	NMIBC: 18 MIBC: 131	Yes	IHC	OS	CD3	0.84	0.68–1.04	0.11	-
						CD8	0.80	0.63–1.02	0.07	
					CSS	CD3	0.81	0.63-1.03	0.089	
						CD8	0.75	0.56-1.01	0.56	
[16]^7^	302	NMIBC: 212 MIBC: 90	Yes	IHC	OS	CD8 ^INT^	0.43	NR	0.003	NR
						CD8 ^ST^	2.21	NR	0.009	
[17]	44	NMIBC: 5 MIBC: 39	No	IHC	CSS	CD8	0.95	0.20–3.55	0.947	T and N stages, and CD169^+^ cells
[26]	67	T1	Yes	IHC	OS	CD8 ^INT^	0.36	0.07–1.40	0.13	Tumor size and multifocality
[9]	221	NMIBC: 43 MIBC: 178	Yes	IHC	OS	Immunoscore	2.01 ^1^	1.20–3.36	0.008	T and N stages, and vascular invasion

Abbreviations: GZMB = granzyme B; IHC = immunohistochemistry; IM = invasive margin, INT = intraepithelial, LVI = lymphovascular invasion, MIBC = muscle-invasive bladder cancer; (N)AC = (neo)adjuvant chemotherapy; NMIBC = nonmuscle invasive bladder cancer; NR: not reported, RFS = recurrence-free survival; RNA-seq = RNA sequencing; ST = stromal; UC: urothelial carcinoma. ^1^ HR > 1 indicates increased risk of an event in patients with higher immune cell infiltration, except for the Immunoscore in the last row. ^2^ OR instead of HR. OR >1 indicates increased risk of recurrence. ^3^ Positive association with CSS (100% vs. 70%). ^4^ Datasets partially overlap. ^5^ Positive association with OS. ^6^ Negative association with OS (median OS 48 vs. 59 months). ^7^ Only an abstract is available.

**Table 2 cancers-12-02692-t002:** Prognostic value of macrophages in BC.

Ref	N	Stage	UC only	Methods	Relation with survival					
					Outcome Parameter	Immune Marker	HR ^1^	95% CI	*p*	Adjusted for
[8]	253	NMBIC: 201 MIBC: 52	Yes	IHC	PFS-NMIBC	CD68	1.52	1.08–2.14	0.016	-
						CD163	NR	NR	NS	
					CSS-MIBC	CD68	3.88	1.00–15.08	0.051	
						CD163	0.99	0.69–1.41	0.95	
						CD68/CD3 > 1	7.73	3.13–19.10	9.5 × 10^-6^	T stage
[42]	94	T1, high grade	Yes	IHC	RFS	CD163	2.11	1.04–4.28	0.038	Gender, age, tumor size, multifocality, CIS, bladder instillations
					PFS	CD163	6.69	1.99–22.50	0.002	
					CSS	CD163	5.86	1.41–24.30	0.015	
[40]	302	NMIBC: 212 MIBC: 90	Yes	IHC	RFS	CD68 ^INT/ST^ CD204 ^INT/ST^ CD169 ^INT/ST^	NR	NR	NS	-
					OS	CD68 ^INT^	1.38	1.88–2.18	0.162	
						CD68 ^ST^	0.91	0.53–1.57	0.735	Age, tumor size, T and N stages, grade
						CD204 ^INT^	1.04	0.59–1.84	0.887	
						CD204 ^ST^	1.981	1.10–3.56	0.022	
						CD169 ^INT^	1.51	0.91–2.52	0.11	
						CD169 ^ST^	1.60	0.94–2.73	0.08	
[17]	44	NMIBC: 5 MIBC: 39	No	IHC	DSS	CD169	0.13	0.01–0.76	0.021	T and N stages, CD8^+^ T cells
[12]	404	MIBC	Yes	RNA-seq, TIMER	OS	TAMs	17.35	5.86–51.33	0.009	Age, stage, CD4^+^ and CD8^+^ T cells, neutrophils, TAMs and DCs
[24]	405	MIBC	Yes	RNA-seq ^6^, CIBERSORT	OS	TAMs	16.57	4.98–55.17	<0.000	-
						M1 TAMs	0.72	0.04–13.68	0.83	
						M2 TAMs	0.94	0.16–5.65	0.95	
	118	MIBC	No	IHC	OS	CD68 ^INT^	1.03	1.01–1.05	0.005	
						CD68 ^ST^	1.02	1.01–1.02	<0.000	
	140	MIBC	No	IHC	OS	CD68 ^INT^	1.04	1.02–1.05	<0.000	
						CD68 ^ST^	1.01	1.01–1.02	<0.000	
[10]	115	Low-grade NMIBC	Yes	IHC	RFS	CD68	3.88 ^2^	1.00–15.08	0.051	Tumor size, stage, GZMB and CD3,4,8,20,56
[43]	141	MIBC	No	IHC	OS	CD68	1.23	0.76–1.99	0.40	T stage, grade, LVI and AC
					RFS	CD68	1.40	0.78–2.52	0.26	
[30]	93	Ta: 69 T1: 24	No	IHC	RFS	CD68	1.19	0.56–2.54	0.65	Gender, smoking status, age, stage, grade and multifocality
[44]	134	NMIBC: 19 MIBC: 115	Yes	IHC	OS	CD163	NR ^3^	NR	0.074	-
					PFS	CD163	NR ^3^	NR	0.090	
[45]	184	NMIBC: 81 MIBC: 11	Yes	IHC	OS	CD68	1.01	0.99–1.01	0.19	Grade, T stage, age and multifocality

Abbreviations: CSS = cancer-specific survival; Gal-9 = Galectin-9; GZMB = granzyme B; IHC = immunohistochemistry; IM = invasive margin; INT = intraepithelial; MFS = metastasis-free survival; MIBC = muscle-invasive bladder cancer; NMIBC = nonmuscle invasive bladder cancer; OS = overall survival; PFS = progression-free survival; RFS = recurrence-free survival; RNA-seq = RNA sequencing; ST = stromal; TAMs = tumor-associated macrophages; UC: urothelial carcinoma. ^1^ HR greater than 1 indicates increased risk of recurrence or death in patients with higher immune cell infiltration. ^2^ OR instead of HR. OR >1 indicates increased risk of recurrence. ^3^ 3. Five-year OS 22% vs. 35.3% in patients with high vs. low CD163^+^ infiltration; Five-year PFS 23.7% vs. 32.2%.

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
