# Peer review of "Prognostic and Predictive Value of Tumor-Infiltrating Immune Cells in Urothelial Cancer of the Bladder"

_cancers, 2020, doi:10.3390/cancers12092692_

Round 1
Reviewer 1 Report
The authors have reviewed the role and presence of immune markers on individual immune cell populations and correlated with clinical outcome. It is well written. However, the nomenclature is confusing. The authors use UC, NMIBC, MIBC throughout the manuscript. These need to be stratified and separated. They are two different diseases, and therefore the distinction needs to be made pretty clearly.
In addition, the correct terminology is intra-epithelial, peritumoral and stromal. This needs to be modified within the manuscript instead of using "tumor center" or "invasive margin".
Author Response
“The authors have reviewed the role and presence of immune markers on individual immune cell populations and correlated with clinical outcome. It is well written. However, the nomenclature is confusing. The authors use UC, NMIBC, MIBC throughout the manuscript. These need to be stratified and separated. They are two different diseases, and therefore the distinction needs to be made pretty clearly.”
Reply: We agree that it is good to make a clear distinction between urothelial cancer and bladder cancer (MIBC/NMIBC). The focus of our review is on urothelial carcinoma of the bladder, which accounts for >90% of bladder cancers. All studies in the prognostic part of the review are about bladder cancer (not upper tract disease). Whereas most studies in the prognostic part of the review focus on urothelial carcinoma of the bladder, some studies did not exclude patients with bladder cancer of other histological variants. In the part of the review that is about checkpoint inhibitors response prediction most studies included patients with urothelial carcinoma, regardless of the location of origin (bladder/ureter/urethra/renal pelvis).
To be clear and correct we made a couple of changes:
-
- We have added a few lines of text in the introduction to clarify the nomenclature
- In the prognostic part, we now use BC/NMIBC/MIBC. In the two tables summarizing the data on T cells and macrophages, we have added an extra column to indicate whether the study only included patients with UC histology or also other histological variants.
- In the predictive part, we use UC and added an extra line of text to clarify this.
“In addition, the correct terminology is intra-epithelial, peritumoral and stromal. This needs to be modified within the manuscript instead of using "tumor center" or "invasive margin".”
Reply: We have changed tumor center into intra-epithelial. However, we prefer to use invasive margin instead of peritumoral as 1) the studies we discuss also use this terminology, 2) we feel that “invasive margin” is a commonly used, well-defined term (it is the region centered at the border of a tumor nest, including a small stromal and a small intra-epithelial region). Although peritumoral is also sometimes used to describe this region, it can also be used to describe only the stromal component of the invasive margin (e.g. https://doi.org/10.18632/oncotarget.14736). We added a description of the invasive margin in the figure legend to clarify the terminology.
Reviewer 2 Report
I think that this review well summarized prognostic and predictive value of tumor-infiltrating immune cells in urothelial cancer.
My comments are below:
Abbreviation of TAAs should be shown in Introduction.
Prognostic and predictive value of eosinophils should be added. It is known that eosinophils play crucial roles for anti-tumor immunity in some cancer.
Prognostic value of mast cells should be added if there are some previous reports showing the association.
Author Response
“Abbreviation of TAAs should be shown in Introduction.”
Reply: We have added this abbreviation in the introduction
“Prognostic and predictive value of eosinophils should be added. It is known that eosinophils play crucial roles for anti-tumor immunity in some cancer.
Prognostic value of mast cells should be added if there are some previous reports showing the association.”
Reply: There is not a lot data on the prognostic value of eosinophils and mast cells in bladder cancer, but we agree that it is good to add this. Therefore, we have added two short paragraphs on the prognostic value of eosinophils and mast cells in BC. There is no data on the predictive value of eosinophils for response to checkpoint inhibitors in BC.
Reviewer 3 Report
The paper describes tumor infiltrating immune cells and immune checkpoint molecules that can be used for predicting prognosis and treatment outcome. The article provides a useful review of IHC studies on urothelial cancer, and also supporting data on RNA-sequencing. The described information will be of use to clinician and researchers involved in the study of urothelial cancer.
I only have the following minor concerns,
- Page 1: An abbreviation of tumor-associated antigen (TAA) should be provided on the line 37.
- Page 3: Authors classified tumors into three immune phenotypes. I suggest that the authors describe a typical histological appearance of each immune phenotypes either in the main text or in the Fig. 1 legend.
Author Response
“1. Page 1: An abbreviation of tumor-associated antigen (TAA) should be provided on the line 37.”
Reply: We have added this abbreviation in the introduction
“2. Page 3: Authors classified tumors into three immune phenotypes. I suggest that the authors describe a typical histological appearance of each immune phenotypes either in the main text or in the Fig. 1 legend.”
Reply: We have a description of the histological appearance of the three immune phenotypes
Reviewer 4 Report
This review is very well written and gives a comprehensive overview of the topic. I have only made som smaller changes to the text (please find attached pdf) with highlighted sections and comments. I think one clarification of a conclusion in the abstract could be made.
The authors write "While PD-L1 expression appears unsuccessful as biomarker for response to checkpoint 
inhibitors, there are some indications that CD8+ T cell infiltration, transforming growth factor-beta 
and myeloid-derived suppressor cells are associated with response. 
” It is not completely clear for the reader if it is increases or decreases and what applies to each of these assessments. I would suggest that the authors rephrase to something similar to they way this is summarized in the conclusions where this is made very clear. “So far, no biomarker has been identified that can accurately predict response to immune 
checkpoint inhibitors in UC. However, high PD-L1 expression, high CD8+ T cell infiltration, low TGF-β signaling and low MDSCs have been associated with higher response rates to checkpoint inhibitors. 

Author Response
“I have only made som smaller changes to the text (please find attached pdf) with highlighted sections and comments. I think one clarification of a conclusion in the abstract could be made.”
Reply: We have accepted all the small changes in the text
“The authors write "While PD-L1 expression appears unsuccessful as biomarker for response to checkpoint 
inhibitors, there are some indications that CD8+ T cell infiltration, transforming growth factor-beta 
and myeloid-derived suppressor cells are associated with response. 
” It is not completely clear for the reader if it is increases or decreases and what applies to each of these assessments. I would suggest that the authors rephrase to something similar to they way this is summarized in the conclusions where this is made very clear. “So far, no biomarker has been identified that can accurately predict response to immune 
checkpoint inhibitors in UC. However, high PD-L1 expression, high CD8+ T cell infiltration, low TGF-β signaling and low MDSCs have been associated with higher response rates to checkpoint inhibitors.”
Reply: We think this is a good suggestion and changed the text in the abstract accordingly.